# Measuring Mutual Policy Divergence for Multi-Agent Sequential Exploration

**Haowen Dou**[1,2,3]  **Lujuan Dang**[1,2,3,⋆]  **Zhirong Luan**[4]

**Badong Chen**[1,2,3,⋆]

[1]National Key Laboratory of Human-Machine Hybrid Augmented Intelligence,
[2]National Engineering Research Center for Visual Information and Applications,
[3]Institute of Artificial Intelligence and Robotics, Xi'an Jiaotong University,
[4]School of Electrical Engineering, Xi'an University of Technology
douhaowen@stu.xjtu.edu.cn, danglj@xjtu.edu.cn, luanzhirong@xaut.edu.cn
chenbd@mail.xjtu.edu.cn

## Abstract

Despite the success of Multi-Agent Reinforcement Learning (MARL) algorithms in cooperative tasks, previous works, unfortunately, face challenges in heterogeneous scenarios since they simply disable parameter sharing for agent specialization. Sequential updating scheme was thus proposed, naturally diversifying agents by encouraging agents to learn from preceding ones. However, the exploration strategy in sequential scheme has not been investigated. Benefiting from updating one-by-one, agents have the access to the information from preceding agents. Thus, in this work, we propose to exploit the preceding information to enhance exploration and heterogeneity sequentially. We present Multi-Agent Divergence Policy Optimization (MADPO), equipped with mutual policy divergence maximization framework. We quantify the discrepancies between episodes to enhance exploration and between agents to heterogenize agents, termed intra-agent divergence and inter-agent divergence. To address the issue that traditional divergence measurements lack stability and directionality, we propose to employ the conditional Cauchy-Schwarz divergence to provide entropy-guided exploration incentives. Extensive experiments show that the proposed method outperforms state-of-the-art sequential updating approaches in two challenging multi-agent tasks with various heterogeneous scenarios. Source code is available at `https://github.com/hwdou6677/MADPO`.

## 1 Introduction

Multi-Agent Reinforcement Learning (MARL) plays an increasingly important role in numerous real-world cooperative problems, such as smart grid management [Zhang et al., 2022b], autonomous driving [Wang et al., 2023b], unmanned system control [Feng et al., 2023b] and games [Zhang et al., 2022a]. Centralized and decentralized MARL methods have been investigated as the first two extensions from single-agent to multi-agent systems. However, challenges have arisen regarding the curse of dimensionality and non-stationary training as the number of agents increasing [Mao et al., 2022]. To address this issue, Centralized Training with Decentralized Execution (CTDE) was developed to disentangle training and execution phases [Foerster et al., 2018]. In the CTDE scheme,

---

⋆ Corresponding authors.

38th Conference on Neural Information Processing Systems (NeurIPS 2024).

the centralized critic provides global information, guiding agents during training but not during execution. CTDE significantly simplifies and stabilizes the training process, providing an effective and efficient paradigm for policy-gradient cooperative MARL.

In CTDE, agents share parameters for homogeneous tasks, such as multi-particle coordination, and then take actions sampled from the same policies. For heterogeneous tasks, such as multi-joint coordination in robotic control, they learn distinct policies without sharing parameters and exhibit different behaviors. However, in these scenarios, relying solely on the non-parameter-sharing setting to achieve cooperation is an oversimplification [Bhattacharya et al., 2023]. This is because agents can never learn optimal policies that depend on trajectories from other agents when updating simultaneously. To tackle this problem, sequential updating [Bertsekas, 2021] has been proposed to improve heterogeneity and collaboration. This updating scheme originates from the insight that agents in one rollout update their policies one-by-one, rather than simultaneously, to retain preceding agent information. Several sequential methods have been proposed by leveraging the multi-agent advantage decomposition lemma[Kuba et al., 2022], the multi-agent performance difference lemma [Wang et al., 2023a], and rollout policy iteration [Bertsekas, 2021], to not only adapt the sequential updating scheme but also maintain the monotonic improvement property.

Despite the success of sequential policy updating, the exploration towards further heterogeneity improvement remains unexplored and challenging [Zhang et al., 2022a]. In MARL, agents struggle to learn globally optimal policy due to the huge exploration space complexity, which is, unfortunately, further amplified in heterogeneous tasks. Existing multi-agent exploration strategies typically require parameter sharing in homogeneous scenarios. However, when applied to heterogeneous scenarios, they suffer from performance degeneration despite employing the non-parameter sharing setting. This is because these methods fail to fully leverage the main advantage of sequential updating, *i.e.* the preceding information. To the best of our knowledge, there is no exploration method that can adapt to both heterogeneous scenarios with sequential updating and homogeneous scenarios with simultaneous updating.

To this end, this paper presents a novel sequential MARL framework, termed **M**ulti-**A**gent **D**ivergence **P**olicy **O**ptimization (MADPO), where a simple yet efficient exploration strategy is equipped to enhance sample efficiency, particularly in heterogeneous scenarios. In MADPO, we first propose a Mutual Policy Divergence Maximization (Mutual PDM) strategy to heterogenize agents. Specifically, mutual PDM consists of the intra-agent divergence and the inter-agent divergence. The intra-agent divergence measures the policy discrepancy between episodes, encouraging agents to learn diversified policies. The inter-agent divergence measures the policy discrepancy between agents, enhancing heterogeneity and promoting greater diversity. However, simply applying classical divergence measures to the proposed framework may trap the exploration in local optima due to the lack of positive incentives. To address this issue, we propose to employ conditional Cauchy-Schwarz (CS) divergence to provide entropy-guided incentives. Compared to the famous Kullback-Leibler (KL) divergence, the conditional CS divergence implicitly maximizes the entropy of current policy and is more stable. The main contributions can be summarized as follows:

1. We develop a novel multi-agent divergence reinforcement learning model equipped with mutual policy divergence maximization, termed MADPO, to enhance exploration and heterogenize agents in heterogeneous scenarios. To the best of our knowledge, we are the first to demonstrate the efficacy of policy divergence maximization in sequential MARL.

2. We propose to maximize conditional Cauchy-Shwarz policy divergence to provide entropy-guided incentive and stabilize multi-agent sequential exploration.

3. We evaluate the proposed method through extensive experiments. The results show that MADPO outperforms state-of-the-art sequential methods in two multi-joint coordination tasks with various heterogeneous scenarios.

## 2 Related Works

### 2.1 Multi-Agent Reinforcement Learning with CTDE

Benefiting from the CTDE framework, multi-agent policy gradient algorithms have paved a promising path for cooperative games [Chai et al., 2021, Qiu et al., 2021, Li et al., 2021]. For example, Wu et al. [2021] proposed CoPPO which guarantees the joint policy improvement by adapting the step size

dynamically. Yu et al. [2022] proposed Multi-Agent Proximal Policy Optimization (MAPPO) which applies PPO to multi-agent scenarios without violating the guarantee of monotonic improvement in the individual policy level. Policy entropy incentive in MAPPO is one of the most related parts to our method, providing diversified policy learning from an information theory perspective. Li and He [2023] proposed MATRPO to extend Trust Region Policy Optimization (TRPO) to multi-agent tasks through a fully decentralized setting and distributed optimization. However, when the number of agents becomes large, MATRPO may encounter the challenge of the connecting link dimension curse. This is because it relies on communication rather than global information to facilitate cooperation. Guo et al. [2024] proposed MASPG, a trust region-based MARL algorithm in the off-policy manner, to enhance the sample efficiency of trust region methods. However, these methods require homogeneity of agents, *i.e.* parameter sharing, to ensure monotonic improvement. This homogeneity assumption can impose significant restrictions on agents, limiting their ability to explore the joint policy space adequately [Ding et al., 2022]. Consequently, if the sharing of parameters is canceled, it can lead to violations of the monotonicity guarantee and result in performance degradation [Zhan et al., 2023].

## 2.2 Sequential Updating MARL

The sequential updating scheme originates from single-agent rollout and policy iteration [Bertsekas, 2021], aiming to update policies of agents one by one, as shown in Fig. 1a. This structure encourages agents to learn different policies based on information from preceding ones, thereby naturally generalizing the homogeneous MARL to heterogeneous MARL. To build the multi-agent sequential updating scheme, attempts have been addressed from both joint and individual policy perspectives. For example, Bertsekas [2021] proposed rollout and policy iteration method, which was the first to consider sequential updating in MARL. Kuba et al. [2022] observed the multi-agent advantage decomposition lemma and proposed HAPPO. Leveraging this powerful lemma, HAPPO estimates and decomposes the joint advantage function to implement sequential updating and ensure the joint monotonic improvement. On the other hand, A2PO proposed by Wang et al. [2023a] focuses on individual policy improvement by leveraging the multi-agent policy difference lemma. A2PO maintains distribution invariance during each agent's advantage estimation process and consider a more refined updating order. Zhao et al. [2023] introduced a localized action value function as the surrogate optimization objective, offering a provable convergence guarantee for multi-agent PPO.

## 2.3 Information Theory Induced RL

Information-theoretic principles serve as a powerful regularization technique for providing valuable guidance in intrinsic reward-driven RL [Liu and Zhang, 2023, Subramanian et al., 2022, Russo and Proutiere, 2024], including both policy and state exploration [Cen et al., 2021, Jacob et al., 2022]. For example, the Soft Actor-Critic (SAC) is the first to maximize the Shannon entropy of policies, promoting randomness and encouraging exploration [Haarnoja et al., 2018]. On-policy methods, such as PPO, MAPPO and HAPPO, embrace the same concept by incorporating entropy regularization into the optimization objective. Additionally, recent advancements have explored the utilization of various entropy forms, such as encoder estimated stable entropy [Liu and Abbeel, 2021], value conditional entropy [Kim et al., 2023] and Rényi entropy [Yuan et al., 2023] to model environmental dynamics and accelerate novel state discovery. However, maximizing entropy only introduces stochasticity for measuring uncertain dynamics. To address this limitation, policy divergence regularization between episodes [Su and Lu, 2022, Xu et al., 2023] has been proposed. This regularization method calculates the policy divergence based on a fixed policy and offers more directed guidance compared to entropy alone. Furthermore, the efficacy of state divergence in combating local optima and fostering state novelty has been demonstrated [Hong et al., 2018, Yang et al., 2021]. However, the existing divergence RL methods cannot be effectively extended to the sequential updating paradigm.

In this work, we pursue an on-policy method to enhance exploration and heterogenize agents in a sequential updating paradigm, termed **M**ulti-**A**gent **D**ivergence **P**olicy **O**ptimization (MADPO). In contrast to the aforementioned policy divergence-based methods, we introduce a novel approach that maximizes inter- and intra-agent policy divergence, thereby incorporating policy information. To further address the deficiency of exploration direction in the traditional divergence RL, we propose to employ the conditional Cauchy-Schwarz divergence to provide an entropy-guided incentive.

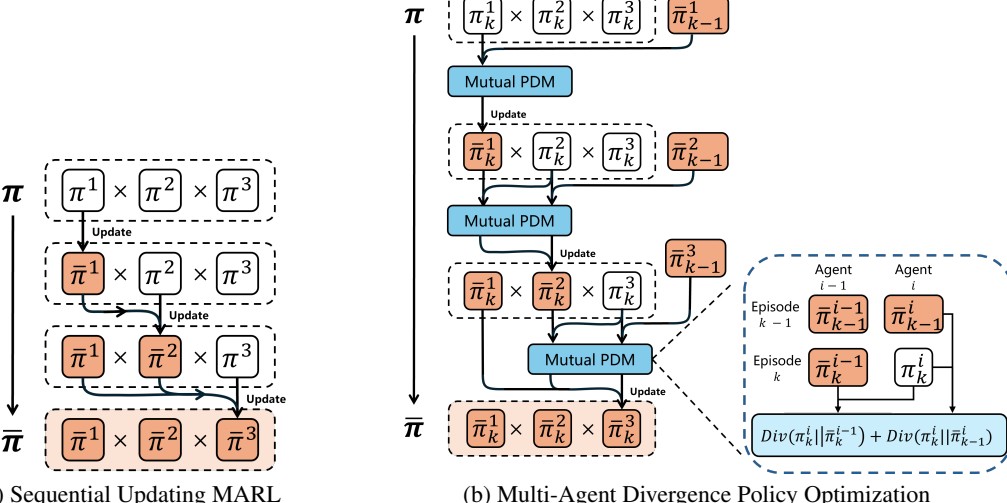

(a) Sequential Updating MARL  (b) Multi-Agent Divergence Policy Optimization

Figure 1: A three-agent example of traditional sequential updating MARL and our MADPO. The white boxes represent the policies to be updated $\pi^i$, and the orange boxes represent the updated policies $\bar{\pi}^i$. The white boxes with dashed lines represent the joint policies to be updated $\boldsymbol{\pi}$, and the orange ones represent the updated joint policy $\bar{\boldsymbol{\pi}}$. Compared to the traditional sequential updating MARL, our method takes the intra-agent and inter-agent divergence into account, as shown in the blue boxes. The intra-agent divergence directs agents to explore novel policies based on their former policies, while the inter-agent divergence heterogenizes agents sequentially.

## 3 Preliminaries

### 3.1 MARL Porblem Formulation

In this paper, we consider a multi-agent sequential decision-making problem, which can be described as a decentralized Markov decision process (DEC-MDP). A DEC-MDP with $n$ agents can be formulated as the tuple: $\langle \mathcal{S}, \boldsymbol{\mathcal{A}}, r, \mathcal{T}, \gamma \rangle$, where $\mathcal{S}$ represents the state space. We denote $N = \{1, \ldots, n\}$ as the set of finite agents. $\boldsymbol{\mathcal{A}} = \mathcal{A}^1 \times \cdots \times \mathcal{A}^n$ is the joint action space by taking the product of actions spaces of $n$ agents. $\mathcal{T} : \mathcal{S} \times \boldsymbol{\mathcal{A}} \times \mathcal{S} \mapsto [0, 1]$ is the state transition function of the environment dynamics. $r : \mathcal{S} \times \boldsymbol{\mathcal{A}} \mapsto \mathbb{R}$ is the reward function. $\gamma$ is the discount factor. At time step $t$, to interact with the environment, each agent at state $s_t \in \mathcal{S}$ takes an action $a_t^i$ from its own policy $\pi^i(\cdot|s_t)$ to form a joint action $\boldsymbol{a}_t = \{a_t^1, \ldots, a_t^n\}$ and a joint policy $\boldsymbol{\pi}(\cdot|s_t) = \pi^1 \times \ldots \times \pi^n$. The agents then receive a joint reward $r(s_t, \boldsymbol{a}_t)$ and step to the new state $s_{t+1}$ with the probability $\mathcal{T}(s_{t+1}|s_t, \boldsymbol{a}_t)$. The objective is to learn an optimal joint policy by maximizing the expected cumulative reward: $\bar{\boldsymbol{\pi}} = \arg\max_{\boldsymbol{\pi}} \sum_{t=0}^{\infty} \mathbb{E}_{s_t \sim \rho_{\boldsymbol{\pi}}, \boldsymbol{a}_t \sim \boldsymbol{\pi}} \left[ \gamma^t r(s_t, \boldsymbol{a}_t) \right]$, where $\rho_{\boldsymbol{\pi}}$ is the marginal state distribution. Following Bellman Equations, the state-action value function and the state function of state $s_t$ are defined as $Q^{\boldsymbol{\pi}}(s_t, \boldsymbol{a}_t) = r(s_t, \boldsymbol{a}_t) + \sum_{i>t}^{\infty} \mathbb{E}_{s_i \sim \rho_{\boldsymbol{\pi}}, \boldsymbol{a}_i \sim \boldsymbol{\pi}} \left[ \gamma^{i-t} r(s_i, \boldsymbol{a}_i) \right]$, and $V^{\boldsymbol{\pi}}(s_t) = \sum_{i>t}^{\infty} \mathbb{E}_{s_i \sim \rho_{\boldsymbol{\pi}}, \boldsymbol{a}_i \sim \boldsymbol{\pi}} \left[ \gamma^{i-t} r(s_i, \boldsymbol{a}_i) | s_0 = s_t \right]$. And the advantage function is defined as $A^{\boldsymbol{\pi}}(s_t, \boldsymbol{a}_t) = Q^{\boldsymbol{\pi}}(s_t, \boldsymbol{a}_t) - V^{\boldsymbol{\pi}}(s_t)$.

### 3.2 Multi-Agent Sequential Policy Updating Paradigm

Sequential updating paradigm was introduced to alleviate homogeneity in multi-agent reinforcement learning. The overview of sequential updating with a three-agent setting is shown in Fig. 1a. For instance, Heterogeneous-Agent Proximal Policy Optimisation (HAPPO) takes preceding agent information into account by employing the multi-agent advantage decomposition lemma and the joint advantage estimator [Kuba et al., 2022]. At episode $k$, agent $m$ in HAPPO maximizes the extrinsic

multi-agent clipping objective as formulated in Eq. 1,

$$r^E = \mathbb{E}_{s \sim \rho_{\boldsymbol{\pi}_{\theta_k}}, \boldsymbol{a} \sim \boldsymbol{\pi}_k} \left[ \min \left( \frac{\pi^{i_m}(a^i|s)}{\bar{\pi}_k^{i_m}(a^i|s)} \right) M^{i_{1:m}}(s, \boldsymbol{a}), clip \left( \frac{\pi^{i_m}(a^i|s)}{\bar{\pi}_k^{i_m}(a^i|s)}, 1 \pm \epsilon \right) M^{i_{1:m}}(s, \boldsymbol{a}) \right],$$

(1)

where $\bar{\pi}_k^{i_m}(a^i|s)$ is the policy of the $m^{th}$ agent updated at episode $k-1$, the superscript of $r^E$ represents *Extrinsic*, and $M^{i_{1:m}}(s, \boldsymbol{a})$ is the joint advantage estimator of the first to $m^{th}$ agents, which is defined as follows,

$$M^{i_{1:m}} = \frac{\bar{\boldsymbol{\pi}}^{i_{1:m-1}}}{\boldsymbol{\pi}^{i_{1:m-1}}} \hat{A}(s, \boldsymbol{a}),$$

(2)

where $\boldsymbol{\pi}^{i_{1:m-1}} = \prod_{p=1}^{m-1} \pi^{i_p}$ is the joint policy of the first to $m^{th}$ agents, and $\hat{A}(s, \boldsymbol{a})$ is an individual advantage estimator, such as Generalized Advantage Estimation (GAE). Additionally, Eq. 1 is also incorporated with an intrinsic reward term, *i.e.* the policy entropy, defined as $r^I = \mathcal{H}(\pi^{i_m}(a^i|s))$.

## 4 Method

### 4.1 Mutual Policy Divergence Maximization

Most existing divergence RL methods only consider state or policy divergence between episodes in a simultaneous updating scheme, which lacks practicality in heterogeneous scenarios. To address this issue, We introduce the main framework of MADPO in this section, *i.e.* Mutual Policy Divergence Maximization (Mutual PDM), and the framework is shown in Fig 1b. Specifically, we consider a mutual intrinsic reward which consists of two types of policy divergence: inter-agent and intra-agent policy divergence. At episode $k$, agent $i$ maximize mutual policy divergence as follows,

$$r^I_{mutual} = \lambda Div(\pi_k^i | \bar{\pi}_k^{i-1}) + (1 - \lambda) Div(\pi_k^i | \bar{\pi}_{k-1}^i),$$

(3)

where $Div(\cdot)$ is one divergence measurement, $\lambda$ is the coefficient to control the influence of the two divergence, and the superscript $I$ represents *Intrinsic*.

The first term in Eq. 3 is the inter-agent policy divergence, quantifying the discrepancy between policies of the current agent and the preceding agent. In heterogeneous tasks, such as multi-joint control in robotics, each agent has its own specialization. Hence, learning diversified policies for different agents is more desirable in these scenarios. By maximizing the inter-agent divergence, agents are provided with a novel optimization direction towards heterogeneity, resulting in significant diversification. Note that in the simultaneous updating manner, the inter-agent divergence maximization becomes theoretically challenging, since these methods lack access to the information from the preceding agents.

The second term in Eq. 3 is the intra-agent policy divergence, which measures the difference between the current policy and the former policy of an agent. The intra-agent policy divergence encourages agents to learn new and diverse policies based on their previous policies. Consequently, we provide agents an optimization direction towards policy novelty, greatly enhancing exploration.

### 4.2 Conditional Cauchy-Schwarz Policy Divergence

To measure the discrepancy between policies, a natural idea is to use the KL-divergence. At episode $k$, agent $i$ optimizes the KL-divergence between the current policy $\pi_k^i$ and a fixed policy $\bar{\pi}$, which can be defined as follows,

$$D_{KL}(\pi_k^i || \bar{\pi}) = \mathbb{E}_{s_j \sim \rho_{\boldsymbol{\pi}_{\theta_k}}, a_j \sim \pi_k^i} \left( \sum_j \pi_k^i(a_j|s_j) \log \frac{\pi_k^i(a_j|s_j)}{\bar{\pi}(a_j|s_j)} \right)$$

(4)

$$= \mathcal{H}(\pi_k^i, \bar{\pi}) - \mathcal{H}(\pi_k^i),$$

(5)

where $\mathcal{H}(\pi_k^i, \bar{\pi})$ is the cross entropy between $\pi_k^i$ and $\bar{\pi}$, $\mathcal{H}(\pi_k^i)$ is the policy entropy. However, optimizing KL-divergence between policies raises problems regarding instability and inhibition of exploration in MARL. Specifically, when approaching 0, the fixed policy $\bar{\pi}(a_j|s_j)$ in Eq. 4 may

lead to uncontrollability and unreliability of the log term, which is common in initialization and converged phrase of MARL. Moreover, the second term in Eq. 5 minimizes the entropy of the current policy, which brings an opposite optimization direction, thus adversely affecting exploration. To address these issues, we introduce Conditional Cauchy-Schwarz Divergence for policy divergence maximization.

The Conditional CS divergence, recently proposed by Yu et al. [2023], is an extension from classic CS divergence for quantifying the discrepancy between two conditional distributions. Formally, given random variable $A$ and $S$ with a finite data set, the CS inequality is defined as follows,

$$\left| \int p(a)q(a)da \right|^2 \leq \int |p(a)|^2 da \int |q(a)|^2 da, \tag{6}$$

where $p(a)$ and $q(a)$ are the probability density functions. By leveraging Eq. 6, we can obtain the CS divergence, defined as $D_{CS} = -\frac{1}{2} \log \frac{(\int p(a)q(a))^2}{\int p^2(a) \int q^2(a)}$. Similarly, we can derive the conditional CS divergence of two policies, *i.e.* the action distributions conditioned by the states, defined as follows,

$$
\begin{aligned}
&D_{CS}(\pi(a|s)||\bar{\pi}(a|s)) \\
&= -\frac{1}{2} \log \frac{\left(\left(\int_{\boldsymbol{S}} \int_{\boldsymbol{A}} \pi(a|s)\bar{\pi}(a|s)d\boldsymbol{\tau}\right)^2}{\left(\int_{\boldsymbol{S}} \int_{\boldsymbol{A}} \bar{\pi}^2(a|s)d\boldsymbol{\tau}\right)\left(\int_{\boldsymbol{S}} \int_{\boldsymbol{A}} \pi^2(a|s)d\boldsymbol{\tau}\right)} \\
&= -2\log \left(\int_{\boldsymbol{S}} \int_{\boldsymbol{A}} \pi(a|s)\bar{\pi}(a|s)d\boldsymbol{\tau}\right) + \log\left(\int_{\boldsymbol{S}} \int_{\boldsymbol{A}} \pi^2(a|s)d\boldsymbol{\tau}\right) + \log\left(\int_{\boldsymbol{S}} \int_{\boldsymbol{A}} \bar{\pi}^2(a|s)d\boldsymbol{\tau}\right).
\end{aligned}
\tag{7}
$$

We then present some desirable properties of Eq 7.

**Proposition 1.** *Given a policy to be updated $\pi$ and a fixed policy $\bar{\pi}$, and $\alpha$-order Rényi policy entropy $\mathcal{H}_\alpha(\pi) = \frac{1}{1-\alpha} \log \int_{\boldsymbol{A}} \pi^\alpha(a|s)da = \frac{1}{1-\alpha}\mathbb{E}_{a\sim\pi} \log \pi^{\alpha-1}(a|s)$, then we have:*

$$\frac{1}{2}\mathcal{H}_2(\pi) + \frac{1}{2}\mathcal{H}_2(\bar{\pi}) \geq D_{CS}\left(\pi|\bar{\pi}\right). \tag{8}$$

Proofs can be found in Appendix A.1. Proposition. 1 is motivated by [Li et al.] and indicates that the CS divergence between distributions is a lower bound of the sum of 2nd-Rényi entropy of distributions. Consequently, in MARL, maximizing the CS divergence between a target policy and a fixed policy behaves like maximizing the 2nd-Rényi entropy of the target policy, which is a generalized form of Shannon policy entropy [Yuan et al., 2023]. In this way, by taking the conditional CS divergence into account, agents are encouraged to enhance their policy entropy while diversifying their policies. Thus, maximizing the CS policy divergence can provide agents with 2nd-Rényi entropy-guided exploration incentives.

**Proposition 2.** *Given a policy to be updated $\pi$ and a fixed policy $\bar{\pi}$ with a finite action set $\boldsymbol{A} = \{a_0, ..., a_n\}$ at state $s$, then the CS divergence is lower bounded by:*

$$D_{CS}(\pi||\bar{\pi}) \geq -\log n. \tag{9}$$

Proofs can be found in Appendix A.2. Recall that in Eq. 4, it is obvious that the KL-divergence is unstable when the probability of one action approaching 0, a common occurrence in MARL. In contrast, Proposition. 2 demonstrates that the CS divergence has a deterministic lower bound unless the number of actions approaches infinity, which is not feasible in practical MARL. Even if in continuous action tasks, the trajectories sampled from policies have finite actions. Thus, maximizing the CS divergence can provide a more stable guidance for policy optimization.

### 4.3 Multi-Agent Divergence Policy Optimization

We first present the overall optimization objective of MADPO in this section. At episode $k$, agent $i$ in MADPO maximizes the practical objective as follows,

$$\mathcal{J}(\pi^i(a^i|s)) = r^E(\pi_k^i(a^i|s)) + \frac{\lambda}{\sigma}\hat{D}_{CS}(\pi_k^i||\bar{\pi}_k^{i-1};\sigma) + \frac{1-\lambda}{\sigma}\hat{D}_{CS}(\pi_k^i||\bar{\pi}_{k-1}^i;\sigma), \tag{10}$$

where $\hat{D}_{CS}(\cdot||\cdot)$ is the estimator of the conditional CS divergence, and $\sigma$ is the parameter of the estimator. Given trajectories $\boldsymbol{\tau}^{\bar{\pi}} = \{s_1^{\bar{\pi}}, a_1^{\bar{\pi}}, ..., s_n^{\bar{\pi}}, a_n^{\bar{\pi}}\}$ and $\boldsymbol{\tau}^{\pi} = \{s_0^{\pi}, a_0^{\pi}, ..., s_n^{\pi}, a_n^{\pi}\}$ sampled from a fixed policy $\bar{\pi}$ and the current policy $\pi$. The empirical estimator of Eq. 7 can be formulated by using kernel density estimation:

$$\hat{D}_{CS}(\pi(a|s)||\bar{\pi}(a|s)) = \log \left( \sum_{i=1}^{n} \left( \frac{\sum_{j=1}^{n} \mathbf{S}_{ij}^{\bar{\pi}} \mathbf{A}_{ij}^{\bar{\pi}}}{\left(\sum_{j=1}^{n} \mathbf{S}_{ij}^{\bar{\pi}}\right)^2} \right) \right) + \log \left( \sum_{i=1}^{n} \left( \frac{\sum_{j=1}^{n} \mathbf{S}_{ij}^{\pi} \mathbf{A}_{ij}^{\pi}}{\left(\sum_{j=1}^{n} \mathbf{S}_{ij}^{\pi}\right)^2} \right) \right)$$

$$- \log \left( \sum_{i=1}^{n} \left( \frac{\sum_{j=1}^{n} \mathbf{S}_{ij}^{\bar{\pi} \to \pi} \mathbf{A}_{ij}^{\bar{\pi} \to \pi}}{\sum_{j=1}^{n} \mathbf{S}_{ij}^{\pi} \sum_{j=1}^{n} \mathbf{S}_{ij}^{\bar{\pi} \to \pi}} \right) \right) - \log \left( \sum_{i=1}^{n} \left( \frac{\sum_{j=1}^{n} \mathbf{S}_{ij}^{\pi \to \bar{\pi}} \mathbf{A}_{ij}^{\pi \to \bar{\pi}}}{\sum_{j=1}^{n} \mathbf{S}_{ij}^{\pi} \sum_{j=1}^{n} \mathbf{S}_{ij}^{\pi \to \bar{\pi}}} \right) \right), \quad (11)$$

where $\mathbf{S}^{\pi}$ and $\mathbf{A}^{\pi}$ represent the Gram matrices of states and actions sampled from the policy $\pi$: $\mathbf{S}_{ij}^{\pi} = \kappa(s_i^{\pi} - s_j^{\pi})$, $\mathbf{A}_{ij}^{\pi} = \kappa(a_i^{\pi} - a_j^{\pi})$, where $\kappa(\cdot)$ is a Gaussian kernel denoted as $\kappa(\cdot) = \exp(-\frac{||\cdot||^2}{2\sigma^2})$, and $\sigma$ is the parameter of $\kappa(\cdot)$. Moreover, $\mathbf{S}^{\pi \to \bar{\pi}}$ and $\mathbf{A}^{\pi \to \bar{\pi}}$ represent the Gram matrices from distribution (*i.e.* policy) $\pi$ to distribution $\bar{\pi}$, formulated as $\mathbf{S}_{ij}^{\pi \to \bar{\pi}} = \kappa(s_i^{\pi} - s_j^{\bar{\pi}})$. Detailed proofs can be found in Yu et al. [2023].

In contrast to existing sequential methods, starting from the second agent in the first episode, MADPO maintains the buffer data for more time. Specifically, for mutual policy divergence maximization, when finished training in episode $k$, MADPO maintains the minibatch of the updated $k$-th policies for one more episode. We summarize the whole algorithm in Algo. 1.

---

**Algorithm 1** Multi-Agent Divergence Policy Optimization

---

**Input:** Initial joint policy $\boldsymbol{\pi}_0 = \pi_0^1 \times ... \times \pi_0^n$, parameters of Mutual PDM, $\sigma$ and $\lambda$.
1: **for** iteration $k = 1, ..., K$ **do**
2:     Collection trajectories $\mathbf{T}_k = \{\boldsymbol{\tau}_k^1, ..., \boldsymbol{\tau}_k^n\}$ by running $\bar{\boldsymbol{\pi}}_k = \bar{\pi}_k^1 \times ... \times \bar{\pi}_k^n$.
3:     Restore $\mathbf{T}_k$ into the buffer.
4:     Compute the advantage $\hat{A}(s, \boldsymbol{a})$ by using the V network.
5:     **for** agent $i = 1, ..., n$ **do**
6:       **if** not $i = 1$ **then**
7:         Compute the inter-agent divergence $\frac{1-\lambda}{\sigma} \hat{D}_{CS}(\pi_k^i || \bar{\pi}_k^{i-1}; \sigma)$ via trajectories $\boldsymbol{\tau}_k^{i-1}$, $\boldsymbol{\tau}_k^i$ and Eq. 11.
8:       **end if**
9:       Compute the intra-agent divergence $\frac{\lambda}{\sigma} \hat{D}_{CS}(\pi_k^i || \bar{\pi}_{k-1}^i; \sigma)$ via trajectories $\boldsymbol{\tau}_{k-1}^i$, $\boldsymbol{\tau}_k^i$ and Eq. 11.
10:       Compute $\mathcal{J} = r^E + r_{mutual}^I$ and update the actor network by maximizing Eq. 10.
11:       Compute the joint advantage via Eq. 2.
12:     **end for**
13:     Update the V network.
14:     Delete $\mathbf{T}_{k-1}$ from the buffer.
15: **end for**

---

## 5 Experiments

We evaluate the proposed MADPO on two challenging multi-agent heterogeneous environments, **Multi-Agent Mujoco (MA-Mujoco)** [de Witt et al., 2020] and **Bi-DexHands** [Chen et al., 2022]. Multi-Agent Mujoco is a complex and widely used task which necessitates up to 17 different joints of one robot to coordinate for human-like behavior imitation, such as running and walking. Bi-DexHands is a bimanual dexterous manipulation environment, where agents are in control of fingers, hands or joints. Sub-scenarios in Bi-DexHands require agents to collaborate for more complex bimanual tasks, such as opening a door inward and outward, passing an item from one hand to another. We compare MADPO with state-of-the-art MARL algorithms, including one simultaneous method MAPPO [Yu et al., 2022] and sequential methods, such as HATRPO [Kuba et al., 2022], HAPPO [Kuba et al., 2022] and A2PO[Wang et al., 2023a]. Clearly, different agents in the two benchmarks should learn diversified policies. Hence, we switch off the parameter sharing setting for HAPPO, HATRPO, A2PO and our MADPO, and keep sharing parameter in MAPPO. In this work, we conduct experiments of 5 random seeds on 10 scenarios of MA-Mujoco and 10 scenarios of Bi-DexHands.

We also conduct statistical testing experiments by using **rliable** [Agarwal et al., 2021]. Since the environments we used in this work do not have a round end score, we choose the aggregate interquartile mean (IQM) sample efficiency test of rliable for evaluation. The interquartile mean (IQM) computes the mean scores of the middle $50\%$ runs, while discarding the bottom and top $25\%$. Here, we evaluate the performance across multiple tasks, and the total number of runs is $n \times m$, where $n$ is the number of trials for one task, and $n = 5$ in this paper. $m$ is the number of tasks. IQM test is more robust than the mean and has less bias than the median. The experimental details can be found in Appendix B.

## 5.1 Results on MA-Mujoco and Bi-DexHands

Fig. 2 and Fig. 3 show the results on MA-Mujoco and Bi-DexHands. The shaded areas represent the $95\%$ confidence interval. we observe that MADPO consistently outperforms all baselines in MA-Mujoco, especially when the number of agents is large, indicating its efficiency in highly complex scenarios. Additionally, MADPO shows superior in challenging bimanual coordination tasks in Bi-DexHands, while other methods like HAPPO suffer from local optima due to insufficient exploration.

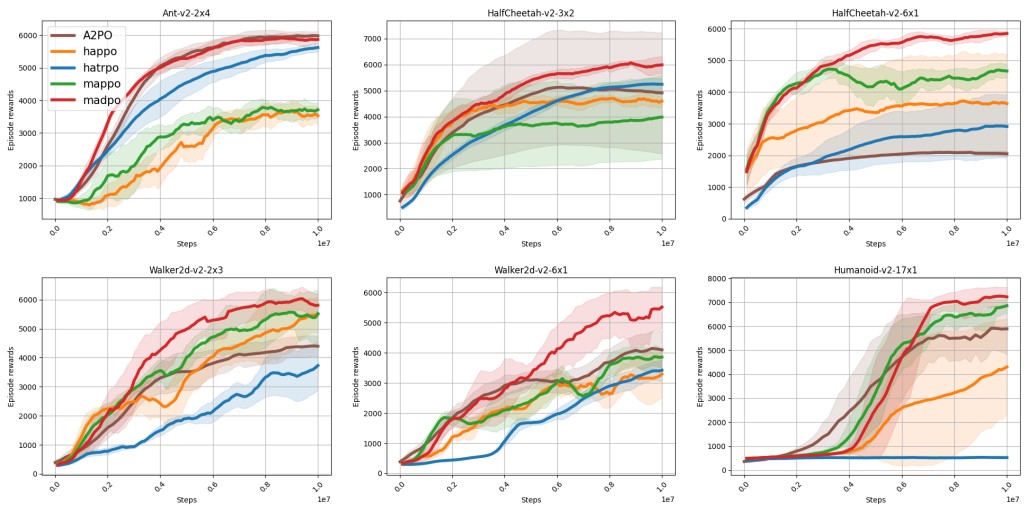

Figure 2: Performance comparison against baseline methods on Multi-Agent Mujoco. Benefiting from the heterogeneity and exploration enhanced by mutual policy divergence maximization, the proposed MADPO consistently outperforms all baselines.

Fig. 4 shows the IQM rewards comparison against other baselines. The lines in the figures represent the IQM, while the shaded areas indicate the confidence intervals. The *10 tasks in MA-Mujoco* include all the tasks of MA-Mujoco used and *10 tasks in Bi-Dexhands* include all the tasks of Bi-Dexhands used. The *3 tasks of MA-Mujoco Ant* include Ant-v2-2x4, 4x2, and 8x1. The *3 tasks of MA-Mujoco Halfcheetah* include Halfcheetah-v2-2x3, 3x2, and 6x1. The *3 tasks of MA-Mujoco Walker2d* include Walker2d-v2-2x3, 3x2, and 6x1. The results in Fig 4 indicates that the proposed MADPO consistently outperforms the state-of-the-art MARL methods in terms of best episodic reward across multiple tasks. The results also show that, MADPO has higher sample efficiency compared to other methods and achieves an improvement gap in most tasks.

## 5.2 Ablation Study

We also investigate the efficiency of conditional CS policy divergence compared to other widely used exploration incentives as shown in Fig. 5a and Fig 5b. Here, *no incentive* presents disabling the intrinsic reward for training. We can clearly observe in Fig. 5a that the conditional CS policy divergence and KL-divergence achieve significant improvements compared to the popular policy entropy. These results indicate the effectiveness of mutual PDM framework, which takes the information from preceding agent into account. Additionally, the conditional CS policy divergence outperforms

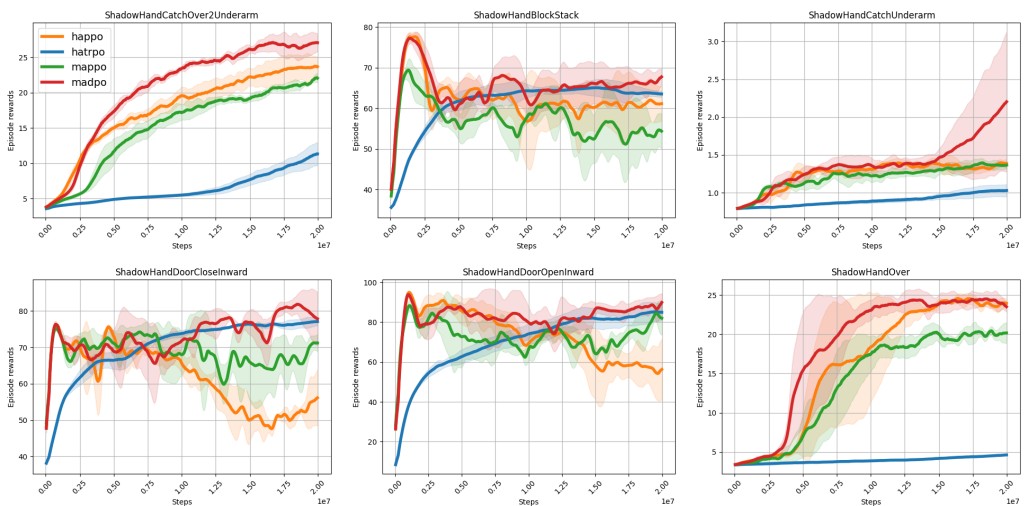

Figure 3: Performance comparison against baseline methods on Bi-DexHands. The proposed MADPO achieves superior performance compared to other MARL methods.

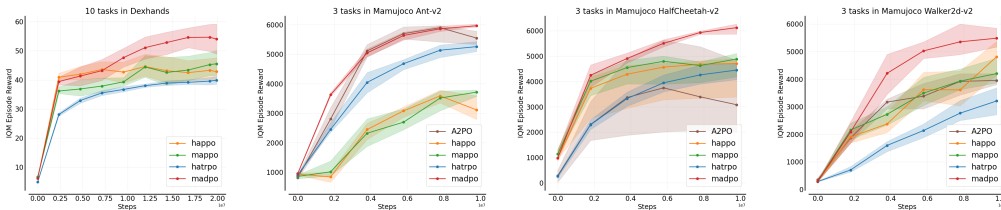

Figure 4: IQM performance comparison against baseline methods on 10 tasks of Bi-DexHands and 9 tasks of MA-mujoco.

the famous KL-divergence empirically, particularly in MA-Mujoco tasks. This is because the KL-divergence implicitly minimizes the entropy of the current policy when maximizing the divergence, which can be detrimental to exploration. In contrast, the CS policy divergence maximizes policies' novelty and, more importantly, the Rényi entropy for efficient exploration. Fig. 5b demonstrates that in the aggregate evaluation, the CS-divergence outperforms other incentives with narrower confidence interval, indicating its better stability than KL-divergence.

Fig. 6 shows the experiments of parameter sensitivity. In this experiment, $\lambda$ controls the influences of inter- and intra-agent policy divergence. When $\lambda = 0$, the inter-agent policy divergence is disabled, and when $\lambda = 1$, the intra-agent policy divergence is disabled. We can observe a consistent degradation in performance when any one aspect of the mutual policy divergence is turned off, thus confirming the significance of our method. When $\lambda = 0.2$, the proposed method achieves the highest reward, whereas excessive influence from inter-agent divergence with $\lambda = 0.5$ is harmful. Parameter $\sigma$ controls the kernel width in Cauchy-Schwarz divergence, impacting the influence of the mutual PDM. We find that MADPO is slightly sensitive to $\sigma$, behaving similarly to the entropy coefficient in MAPPO.

## 6 Conclusion

In this work, we present MADPO, a sequential updating MARL method equipped with mutual policy divergence maximization for efficient exploration in heterogeneous tasks. By leveraging the sequential updating paradigm, MADPO maximizes intra-agent policy divergence to enhance exploration and inter-agent policy divergence to promote heterogeneity. However, maximizing traditional divergence measurements can lead to instability and lack of direction in MARL. To tackle this issue, we propose conditional Cauchy-Schwarz policy divergence to quantify the distance between policies. The

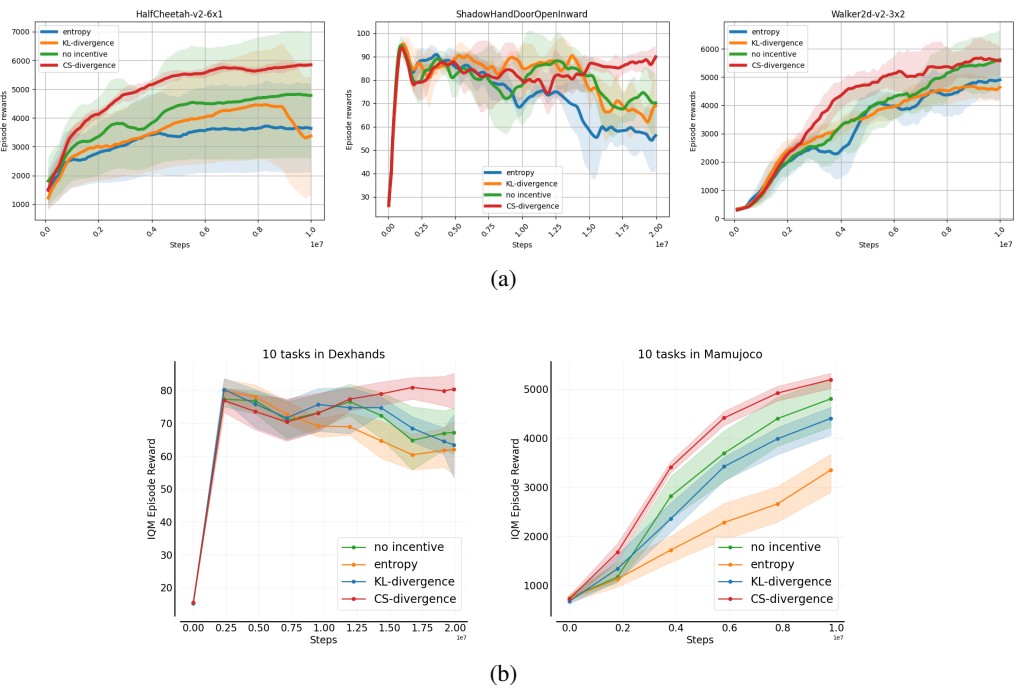

(a)

(b)

Figure 5: Performance comparison against other exploration incentives.

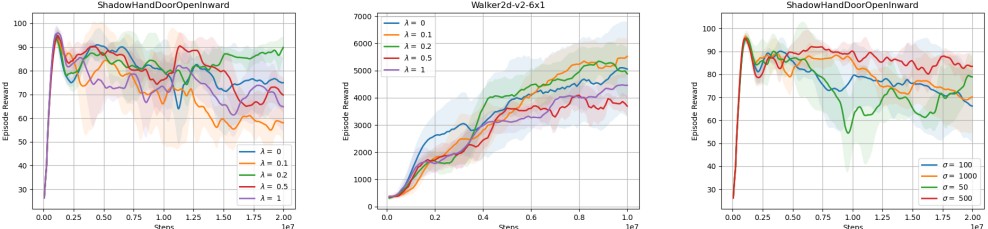

Figure 6: Parameter sensitivity studies for MADPO.

conditional Cauchy-Schwarz policy divergence possesses favorable properties and provides a stable entropy-guided incentive for sequential exploration. We evaluate the performance of MADPO on two challenging heterogeneous tasks, MA-Mujoco and Bi-Dexhands. We observe that the proposed Mutual PDM outperforms entropy-based methods since it consider both previous and preceding information. Moreover, we verify the efficiency of the conditional Cauchy-Schwarz policy divergence in terms of stabilizing and guiding the exploration. Totally, the results demonstrate the effectiveness of MADPO, achieving state-of-the-art performance in complex multi-agent scenarios. The main limitation of this work is that when the number of agent increases, the proposed method may require more ram to restore previous information. We will investigate effective representation methods for previous information in the future.

## Acknowledgments and Disclosure of Funding

This work is supported by the National Key R&D Program of China (2023YFB4704900) and National Natural Science Foundation of China (U21A20485). The authors declare that they have no conflict of interest.

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

## A  Theoretical Analysis

### A.1  Proofs of the Proposition 1

**Proposition 1.** *Given a policy to be updated $\pi$ and a fixed policy $\bar{\pi}$, and their $\alpha$-order Rényi entropy $\mathcal{H}_\alpha(\pi) = \frac{1}{1-\alpha} \log \int_{A} \pi^\alpha(a|s)da = \frac{1}{1-\alpha} \mathbb{E}_{a\sim\pi} \log \pi^{\alpha-1}(a|s)$, then we have:*

$$\frac{1}{2}\mathcal{H}_2(\pi) + \frac{1}{2}\mathcal{H}_2(\bar{\pi}) \geq D_{CS}\left(\pi|\bar{\pi}\right). \tag{12}$$

*Proof.* At state $s$, consider $\boldsymbol{A}$ as the action set, $\boldsymbol{a}$ is the random variable, we can rewrite the right side of Eq. 12 as follows,

$$D_{CS}\left(\pi|\bar{\pi}\right) = -\frac{1}{2} \log \frac{\left(\int_{\boldsymbol{A}} \pi(\boldsymbol{a}=a|s)\bar{\pi}(\boldsymbol{a}=a|s)da\right)^2}{\left(\int_{\boldsymbol{A}} \pi^2(\boldsymbol{a}=a|s)da\right)\left(\int_{\boldsymbol{A}} \bar{\pi}^2(\boldsymbol{a}=a|s)da\right)} \tag{13}$$

$$= -\log\left(\int_{\boldsymbol{A}} \pi(\boldsymbol{a}=a|s)\bar{\pi}(\boldsymbol{a}=a|s)da\right) - \frac{1}{2}\mathcal{H}_2(\pi) - \frac{1}{2}\mathcal{H}_2(\bar{\pi}), \tag{14}$$

where the first term in Eq. 14 is the 2nd-order Rényi cross entropy between $\pi$ and $\bar{\pi}$. Thus, using Gibbs inequality, we have

$$\mathcal{H}_2(\pi) + \mathcal{H}_2(\bar{\pi}) \geq -\log\left(\int_{\boldsymbol{A}} \pi(\boldsymbol{a}=a|s)\bar{\pi}(\boldsymbol{a}=a|s)da\right), \tag{15}$$

$$\frac{1}{2}\mathcal{H}_2(\pi) + \frac{1}{2}\mathcal{H}_2(\bar{\pi}) \geq -\log\left(\int_{\boldsymbol{A}} \pi(\boldsymbol{a}=a|s)\bar{\pi}(\boldsymbol{a}=a|s)da\right) - \frac{1}{2}\mathcal{H}_2(\pi) - \frac{1}{2}\mathcal{H}_2(\bar{\pi}), \tag{16}$$

$$\frac{1}{2}\mathcal{H}_2(\pi) + \frac{1}{2}\mathcal{H}_2(\bar{\pi}) \geq D_{CS}\left(\pi|\bar{\pi}\right), \tag{17}$$

which finish the proof.

## A.2 Proofs of the Proposition 2

**Proposition 2.** *Given a policy to be updated $\pi$ and a fixed policy $\bar{\pi}$ with a finite action set $\boldsymbol{A} = \{s_0, ..., s_n\}$ at state $s$, then the CS divergence of is lower bounded by:*

$$D_{CS}(\pi||\bar{\pi}) \geq -\log n. \tag{18}$$

*Proof.* For two policies represented by the trajectories, the CS divergence between them at state $s$ is defined as follows,

$$D_{CS}(\pi||\bar{\pi}) = -\log\left(\frac{\sum^{\boldsymbol{A}} \pi(\boldsymbol{a} = a|s)\bar{\pi}(\boldsymbol{a} = a|s)}{\sqrt{\sum^{\boldsymbol{A}} \pi(\boldsymbol{a} = a|s)^2}\sqrt{\sum^{\boldsymbol{A}} \bar{\pi}(\boldsymbol{a} = a|s)^2}}\right). \tag{19}$$

Consider the quadratic mean of the $\pi(\boldsymbol{a} = a|s)$,

$$\frac{\sum^{\boldsymbol{A}} \pi(a|s)^2}{n} \geq \left(\frac{\sum^{\boldsymbol{A}} \pi(a|s)}{n}\right)^2 = \frac{1}{n^2}. \tag{20}$$

Hence,

$$\sqrt{\sum^{\boldsymbol{A}} \pi(a|s)^2} \geq \sqrt{\frac{1}{n}}. \tag{21}$$

We also have $\sum^{\boldsymbol{A}} \pi(\boldsymbol{a} = a|s)\bar{\pi}(\boldsymbol{a} = a|s) \leq 1$, and then

$$D_{CS}(\pi||\bar{\pi}) \geq -\log\left(\frac{1}{1/n}\right) = -\log n, \tag{22}$$

which complete the proof.

# B Experimental Results

## B.1 Detailed experimental Settings

Table 1: Common hypermeters in MA-Mujoco

| hyperparameters | MA-Mujoco |
|---|---|
| activation | ReLu |
| batch size | 4000 |
| gamma | 0.99 |
| gain | 0.01 |
| PPO epoch | 5 |
| episode length | 200 |
| n rollout threads | 20 |

Table 2: Different hyperparameters in MA-Mujoco

| Tasks | hidden layer | actor lr | critic lr | clip | $\lambda$ | $\sigma$ |
|---|---|---|---|---|---|---|
| Ant-v2-2x4 | [64,64] | 5e-4 | 5e-4 | 0.2 | 0.5 | 1e3 |
| Ant-v2-4x2 | [64,64] | 5e-4 | 5e-4 | 0.2 | 0.5 | 1e3 |
| Ant-v2-8x1 | [64,64] | 5e-4 | 5e-4 | 0.2 | 0.5 | 1e3 |
| HalfCheetah-v2-2x3 | [64,64] | 5e-4 | 5e-4 | 0.2 | 0.2 | 1e3 |
| HalfCheetah-v2-3x2 | [64,64] | 5e-4 | 5e-4 | 0.2 | 0.2 | 1e3 |
| HalfCheetah-v2-6x1 | [64,64] | 5e-4 | 5e-4 | 0.2 | 0.1 | 1e3 |
| Walker2d-v2-2x3 | [256,256] | 1e-3 | 1e-3 | 0.05 | 0.1 | 2e3 |
| Walker2d-v2-3x2 | [256,256] | 1e-3 | 1e-3 | 0.05 | 0.1 | 2e3 |
| Walker2d-v2-6x1 | [256,256] | 1e-3 | 1e-3 | 0.05 | 0.1 | 2e3 |
| Humanoid-v2-17x1 | [256,256] | 5e-4 | 5e-4 | 0.1 | 0.5 | 1e3 |

Table 3: Common hypermeters in Bi-DexHands

| hyperparameters | BiDexHands |
|---|---|
| activation | ReLu |
| batch size | 4000 |
| gamma | 0.99 |
| gain | 0.01 |
| PPO epoch | 5 |
| episode length | 75 |
| n rollout threads | 128 |
| hidden layers | [256,256,256] |
| clip | 0.2 |
| actor lr | 5e-4 |
| critic lr | 5e-4 |

Table 4: Different hyperparameters in Bi-DexHands

| Tasks | $\lambda$ | $\sigma$ |
|---|---|---|
| ShadowHandBlockStack | 0.2 | 5e2 |
| ShadowHandOver | 0.2 | 5e2 |
| ShadowHandPen | 0.2 | 5e2 |
| ShadowHandDoorCloseInward | 0.5 | 5e2 |
| ShadowHandDoorOpenInward | 0.5 | 5e2 |
| ShadowHandCatchOver2Underarm | 0.2 | 1e3 |
| ShadowHandCatchUnderarm | 0.2 | 1e3 |
| ShadowHandCatchAbreast | 0.2 | 1e3 |
| ShadowHandDoorCloseOutward | 0.5 | 1e3 |
| ShadowHandDoorOpenOutward | 0.5 | 1e3 |

In this experiment, we follow the official implement and hyperparameter settings of HAPPO and HATRPO[1] [Kuba et al., 2022], MAPPO[2] [Yu et al., 2022], and A2PO[3] [Wang et al., 2023a]. We compare the proposed method with baselines on two popular heterogeneous environments, MA-Mujoco[4], and Bi-DexHands[5]. For MA-Mujoco, the commom hyperparameter are listed in Tab. 1, and the different hyperparameters in each scenarios are listed in Tab. 2. For Bi-DexHands, the commom hyperparameter are listed in Tab. 3, and the different hyperparameters in each scenarios are listed in Tab. 4. The experiments were conducted on a PC with NVIDIA RTX3090 GPU, Intel Xeon 64-core CPU, and 64GB Ram.

## B.2 Additional Results

Full results of 10 scenarios in MA-Mujoco and 10 scenarios in Bi-DexHands are shown in Fig. 7 and Fig. 8. We can make two observations on results of MA-Mujoco tasks. First, MADPO demonstrates superiority in terms of both reward maximum and learning speed, highlighting its effectiveness in exploring novel policies. Second, MADPO exhibits the lowest variance compared to other methods in most scenarios, indicating its training stability. In the more challenging Bi-DexHands tasks, we find MADPO outperforms baselines in most scenarios, confirming the effectiveness of MADPO in complex coordination tasks.

We also conduct the experiments of different updating orders in Ma-Mujoco, as indicated in Fig 9. Here, each agent controls one joint of one leg, and the joints in the same position on the legs have the same specilization. The Rand. order represents updating agents randomly. The Def. order represents updating agents in the default order in Ma-Mujoco, where agents are grouped according to the legs

---

[1] https://github.com/PKU-MARL/HARL

[2] https://github.com/marlbenchmark/on-policy

[3] https://github.com/xihuai18/A2PO-ICLR2023

[4] https://github.com/schroederdewitt/multiagent_mujoco

[5] https://github.com/PKU-MARL/DexterousHands

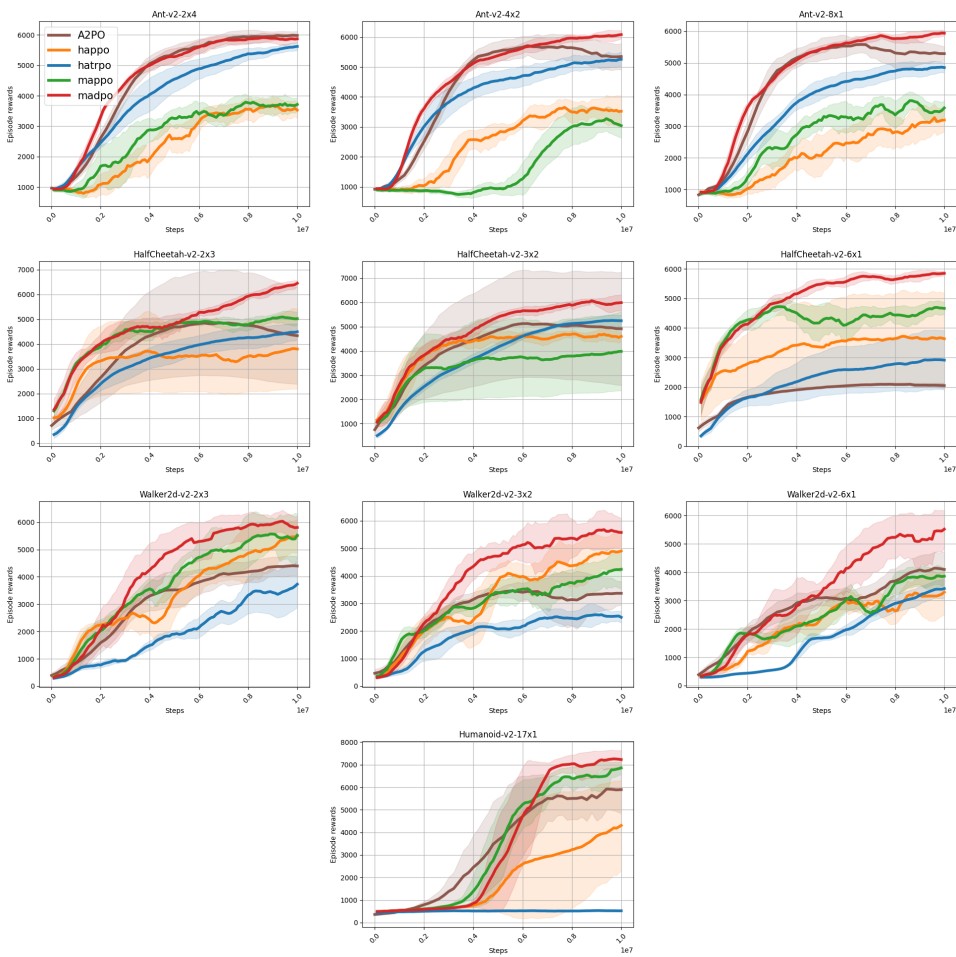

Figure 7: Performance comparison on against baseline methods on 10 Multi-Agent Mujoco scenarios.

they belong to. For example, the default updating order can be: *right thigh joint, right leg joint, right foot joint, left thigh joint, · · · .* The Spec. order represents updating agents according to their specilization, and can be: *right thigh joint, left thigh joint, right foot joint, left foot joint, · · · .* We can observe that the random updating order outperforms the other orders. We believe that this is because our framework can benefit from various updating orders. For example, if the current agent share the same specilization as the preceding agent, maximizing the inter-agent divergence can enhance exploration. On the other hand, if the current agent differs from the preceding agent, maximizing the inter-agent divergence can promote heterogeneity.

In Fig 10, we compare the performance between different parameter settings using IQM aggregate test. We can observe that MADPO is a little sensitive to parameter $\sigma$. However, it outperforms HAPPO in a reasonable range of $\sigma$. Additionally, we can also individually tune $\sigma$ for special task for further performance improvement. We also observe that MAPDO is a little sensitive to the parameter $\lambda$, yet it consistently shows better performance than HAPPO.

Tab. 5 shows the running time of MADPO and other methods. Compared to MARL baselines, MADPO only introduces a negligible extra time cost.

## C   Social Impacts

We do not foresee an obvious negative impart by conducting the experiments included in this work, since we evaluate methods in a controlled simulation environment. However, We are also aware that recent works [Radosavovic et al., 2024, Handa et al., 2023] tested RL algorithms on

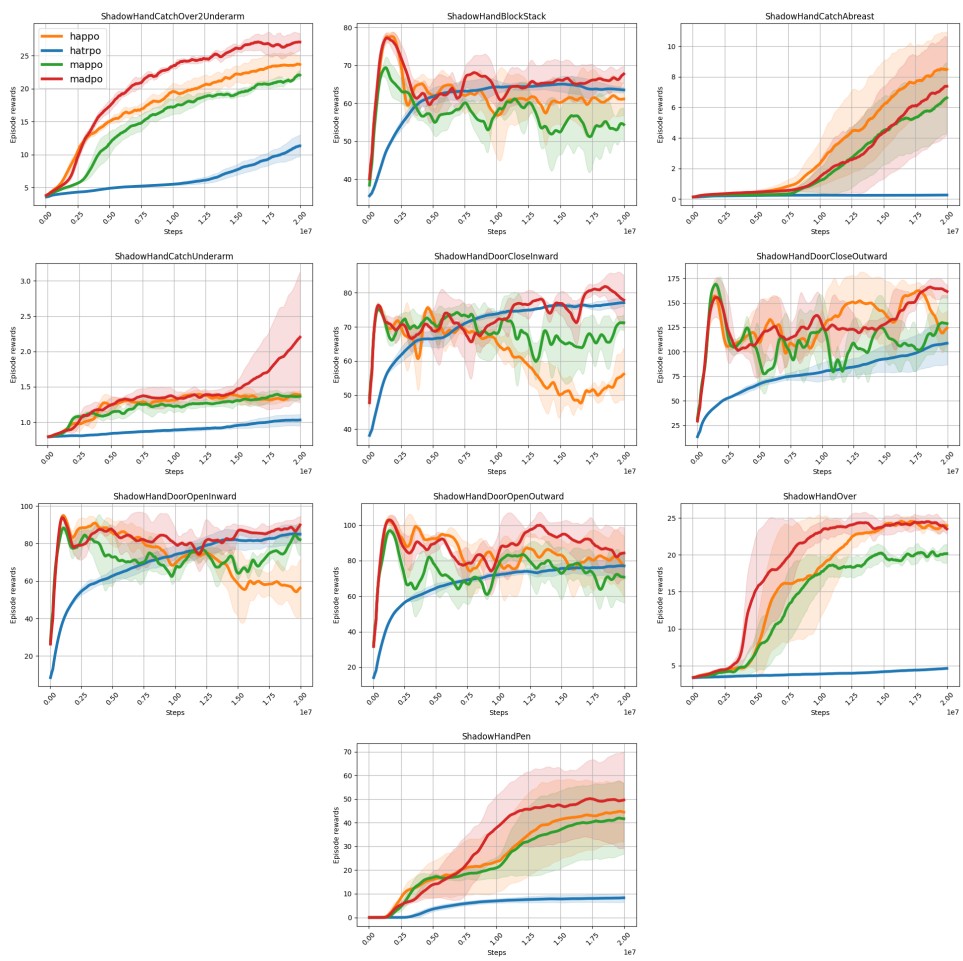

Figure 8: Performance comparison on against baseline methods on 10 Bi-DexHands scenarios.

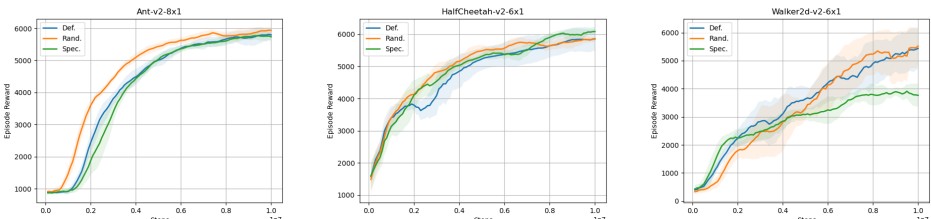

Figure 9: Performance comparison of different updating orders on Ma-Mujoco scenarios.

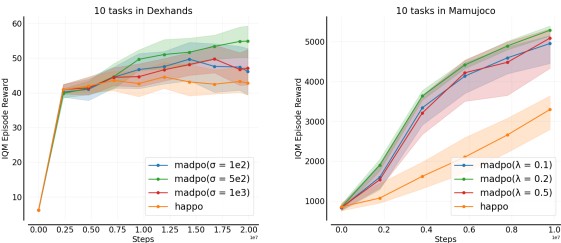

Figure 10: Aggregate parameter sensitivity study.

Table 5: Wall time comparison.

| Task | training steps | A2PO | HAPPO | HATRPO | MAPPO | MADPO |
|------|----------------|------|-------|--------|-------|-------|
| Ant-v2-2x4 | 1e7 | 1h2m | 1h3m | 1h16m | 1h10m | 1h17m |
| Walker2d-v2-6x1 | 1e7 | 1h57m | 2h1m | 2h35m | 1h49m | 2h13m |
| HalfCheetah-v2-6x1 | 1e7 | 2h3m | 1h56m | 2h22m | 1h40m | 2h27m |
| Humanoid-v2-17x1 | 1e7 | 6h20m | 6h8m | 6h51m | 6h16m | 7h3m |
| ShadowHandDoorOpenInward | 2e7 | - | 1h48m | 1h39m | 2h27m | 2h45m |
| ShadowHandDoorOpenOutward | 2e7 | - | 1h50m | 2h4m | 2h7m | 3h12m |

real-world environments. This may raise concerns regarding the potential personal hazards caused by agent exploration in real world. To address this issue, one possible approach is to define safety behaviours to restrict the actions of agents [Feng et al., 2023a] or perform evaluation in safe simulation environments [Ji et al., 2023] preliminarily.

