# OpenReview forum: "Measuring Mutual Policy Divergence for Multi-Agent Sequential Exploration"
_NeurIPS.cc/2024/Conference — NeurIPS 2024 poster_

### Official Review · Reviewer_gpQ4 · 2024-07-11

**Soundness:** 2
**Presentation:** 3
**Contribution:** 2
**Rating:** 6
**Confidence:** 3

**Summary:**

The authors study MARL in heterogeneous settings, where agents are not allowed to share their parameters, and make use of the sequential updating scheme under the CTDE schema. They propose a method which exploits the preceding information to improve exploration and heterogeneity sequentially. This method is equipped with a mutual policy divergence maximization framework, which utilizes the discrepancies between episodes to enhance exploration and between agents to heterogenize agents. Interestingly, the authors propose the conditional Cauchy-Schwarz divergence to provide entropy-guided exploration incentives.

**Strengths:**

- The problem of exploration in settings with heterogeneous agents is important in MARL and not well-explored in literature.

- The paper is the first to study the effectiveness of policy divergence maximization in the sequential updating schema, upon which important related work has been built.

- The paper proposed the conditional Cauchy-Schwarz (CS) divergence as an alternative to the popular KL-divergence in MARL. Such an alternation may be interesting to the broader RL community. Interestingly, unlike KL-divergence which can expload for small values of the denominator, CS divergence has a provable good lower bound ($-\log(n)$) only dependent on the number of finite actions.

- The proposed method displays good performance, in comparison to strong SOTA methods (including MAPPO, HAPPO), on benchmarks with heterogeneous agents.

- The proposed framework is simple and easy-to-implement.

- The paper is generally well-written and easy-to-follow.

**Weaknesses:**

- The improvement over the baselines (standard KL-divergence, entropy term, no incentive) does not seem to be quite consistent in the ablation study, due to (a) high variance in the results of the no incentive, and (b) very close improvement over the KL divergence baseline in terms of best episodic reward in 2 out 3 tasks.

- Since the CS divergence is new in MARL and RL, a table containing the running times of the evaluated algorithms is missing. How costly is the CS divergence?

**Questions:**

- The authors mention: "To the best of our knowledge, there is no exploration method that can adapt to both heterogeneous scenarios with sequential updating and homogeneous scenarios with simultaneous updating". But can the proposed method adapt to homogeneous scenarios with simultaneous updating? No experiments in such settings have been provided. Could the proposed intrinsic rewards be used to improve exploration in MARL settings with homogeneous agents?

- Why do the authors use $\lambda$ and $1 - \lambda$ for weighting the intrinsic rewards, instead of arbitrary weights (not in a convex combination)? How important is it to the performance?

**Limitations:**

The authors provide limitations of their work.

---

> ### Author Rebuttal · Authors · 2024-08-05
>
> Thanks for your positive comments. We address your concerns as follows:
> ## (1) The improvement over baselines
> We have re-evaluated our method against the baselines using an aggregate statistical test. We have quantified the interquartile mean (IQM) across tasks of our method and baselines. Please refer to the author rebuttal and the attached PDF for more details.
>
> In the PDF of the author rebuttal above, Figure 2 (a) and (b)  show the performance comparison with other exploration strategies. We can observe that in terms of IQM reward across multiple tasks, the proposed method consistently outperforms other methods in MA-Mujoco, with small overlap. Besides, the CS-divergence shows more stability and has smaller confidence intervals in MA-Mujoco compared to other methods. In the Bi-Dexhands environment, it shows a significant improvement gap in the last 10m steps and achieves the highest best episodic reward.
>
> ## (2) The running time comparison
> We have compared the running time of our methods and baselines. Please refer to the author rebuttal and the attached PDF for more details.
>
> From Table 1 in the attached PDF in the author rebuttal above, we observe that compared to sequential MARL baselines, MADPO only introduce a negligible extra time cost.
>
> ## (3) Adapt to homogeneous tasks
> Homogeneous scenarios can be regarded as a special case of heterogeneous scenarios. Thus, yes, our method can be adapted to homogeneous scenarios. Existing methods ensure homogeneous agents by enabling parameter sharing. Under such a setting, the intra-agent divergence maximization is unnecessary, since agents should receive a homogeneous intrinsic reward. Consequently, when setting $\lambda$ to $0$, the Mutual PDM can be adapted to homogeneous tasks with simultaneous updating.
>
> Due to the page limit of the attached PDF, we present here the following table, which shows the aggregate IQM best episodic rewards comparison across $10$ tasks and $5$ random seeds in MA-Mujoco, between different exploration strategies when adapted to MAPPO. Here, we disenable the parameter sharing. The results show that as an exploration incentive in MAPPO, the Mutual PDM achieves better performance than the entropy, indicating its effectiveness when adapted to homogeneous agents with simultaneous updating.
> | Steps| MAPPO (entropy)| MAPPO (Mutual PDM with $\lambda =0$)|
> | :---- | :-----------: |:--------:|
> | 2e6| $\pmb{1567.87(\pm138.48)}$|  $1431.23(\pm89.18)$|
> | 4e6| $2139.04(\pm248.98)$|  $\pmb{2494.59(\pm201.06)}$|
> | 6e6| $2594.42(\pm356.38)$|  $\pmb{2954.39(\pm389.27)}$|
> | 8e6| $3032.69(\pm387.55)$|  $\pmb{3228.39(\pm231.90)}$|
> | 1e7| $3577.72(\pm145.59)$|  $\pmb{4155.31(\pm283.54)}$|
>
> ## (4) Why $\lambda$ and $1-\lambda$ for weighting
> We use $\lambda$ and $1-\lambda$ here to balance the two components of mutual policy divergence maximization for the following reasons.
>
> First, it is necessary to scale the two types of divergences to the same range since they share the same range of values. In different tasks with different requirements for heterogeneity and exploration ability, adjusting the degree of the two components is necessary.
>
> Even in tasks requiring heterogeneity and exploration ability both highly, we can tune the other parameter $\sigma$ to adjust the scale of the mutual divergence. Therefore, jointly tuning $\lambda$ and $\sigma$ can achieve the desired parameter combinations, and is sufficient for MADPO to adapt to different tasks. Compared to arbitrary weights, this parameter setting facilitates parameter tuning, since we can use $\sigma$ to control the sacle of mutual divergence and within use $\lambda$ to weighting the two parts.
>
> Additionally, using other weighting methods, such as no-linear weighted mean or boosting learning, may introduce extra parameters and computational costs. Due to the limited time for rebuttal, we will investigate the performance of different weighting methods in the future.

---

> > ### Comment · Reviewer_gpQ4 · 2024-08-09
> >
> > I would like to thank the authors for addressing my concerns and questions. I increased my rating score.

---

> > > ### Author Response · Authors · 2024-08-12
> > > **Thanks for the response**
> > >
> > > Thank you for your response. We are delighted to hear that our rebuttal addressed your concerns well.

---

### Official Review · Reviewer_3yBc · 2024-07-12

**Soundness:** 3
**Presentation:** 3
**Contribution:** 3
**Rating:** 7
**Confidence:** 3

**Summary:**

The paper proposes a novel training objective where it encourages the policies to diverge from each other and from the previous policy under heterogeneous multi-agent tasks based on sequential recently proposed sequential policy update. It utilizes CS divergence for calculation of "distance" between policies for tractable and stable optimization compared to KL divergence. The evaluation is done in high-dimensional multi-agent mujoco and and bi-dexhands environments, outperforming existing state-of-the-art sequential algorithms.

**Strengths:**

- The paper is well written and easy to understand; Fig. 1 is very informative.
- The problem of exploration under agent heterogeneity is an important problem in multi-agent learning
- The proposed method is sound and is backed by theory

**Weaknesses:**

- The evaluation is hard to judge whether the proposed method is actually performs better than the baselines, this is a deal breaker. I suggest the authors also incorporate aggregate quantities from https://agarwl.github.io/rliable/

I'm willing to increase the score if the authors show that the improvement is statistically significant

**Questions:**

- Is it possible to have a "cyclic" problem where the 1st, 3rd, 5th, ... (and also 2nd, 4th, 6th, ...) policies have the same behavior despite optimizing the proposed training objective?
- Can the authors explain why CS is chosen over Jensen–Shannon divergence (JSD)?
- Is there a guideline for tuning the coefficients for the intrinsic rewards?

**Limitations:**

Minor comments
- line 36, wrong citation format
- line 193 and line 217, Ep. 5 should be Eq. 4

---

> ### Author Rebuttal · Authors · 2024-08-05
>
> Thanks for your positive comments. We address your concerns as follows:
> ## (1) The aggregate evaluation metrics
> Thanks for your constructive suggestion. We have re-evaluated our method by using this powerful toolbox ***rliable***. Please refer to the author rebuttal and the attached pdf for more information. Since the environments used in this study does not have a round end score, we choose the aggregate interquartile mean (IQM) across tasks as the evaluation metric. The results in Figure 1 of the PDF demonstrates that, in terms of aggregate IQM, MADPO achieves better performance and has smaller confidence intervals in most tasks. We believe these results has the statistical significance.
> ## (2) A cyclic similarity problem in learned policies
> It is an interesting question, and we believe the probability of such a situation occurring is extremely low. In our MADPO, agents are encouraged to learn different policies from the preceding ones, yet are not to be guided to behave like another one. We admit that there is a chance that one agent can autonomously imitate the one before the preceding one by a divergence incentive, as you mentioned. However, the probability is extremely low due to (a) there is no direct guidance for imitation, (b) agents are also goaded to diversify based on their previous policies (the intra-agent policy maximization), and (c) the CS divergence incentive can bring further stochasticity to agents by implicitly maximizing the policy entropy.
>
> ## (3) Why choose CS divergence over JS divergence
> JS divergence fixes the problem that CS divergence may explode in MARL scenarios, since it has a constant lower bound and upper bound. However, the JS divergence is not the best choice. The JS divergence between the current policy and a fixed policy is defined as follows,
> $$\begin{aligned}
> D_{JS}(\pi||\overline{\pi}) &= \frac{1}{2}D_{KL}(\pi||\frac{\pi+\overline{\pi}}{2}) + \frac{1}{2}D_{KL}(\frac{\pi+\overline{\pi}}{2}||\overline{\pi}) \\\\
> &=\frac{1}{2}[\mathcal{H}(\pi,\frac{\pi+\overline{\pi}}{2})-\mathcal{H}(\pi) + \mathcal{H}(\frac{\pi+\overline{\pi}}{2},\overline{\pi})-\mathcal{H}(\overline{\pi})].
> \end{aligned}$$
> It still has the same drawback as KL divergence, i.e. minimizing the policy entropy $\mathcal{H}(\pi)$ when maximizing the divergence, which is harmful to exploration.
>
> On the contrary, as indicated in Proposition 1, our method implicitly maximizes the policy when maximizing the divergence, which can bring more stochasticity to policies and benefit exploration. Hence, our method is able to provide agents with entropy-guided divergence incentives for exploration.
>
> ## (4) Guideline for tuning parameters
> We have evaluated MADPO under different parameter settings by using the aggregate IQM test of ***rliable***, and thanks again for recommending this method. Please refer to the author rebuttal and the attached PDF above for the experimental results. The results show that our MADPO achieves better performance than HAPPO in a reasonable range of $\sigma$.
>
>  Based on the empirical results, here we can give a guideline for tuning $\sigma$ and $\lambda$: (a) set $\sigma$ as commonly used values, such as 1e2 or 5e2; (b) set $\lambda$ based on the task type, but not more than 0.5.
>
> ## (5) Minor suggestions
> Thanks for pointing out the wrong reference and citation formats, and we have corrected them accordingly.

---

> > ### Comment · Reviewer_3yBc · 2024-08-10
> >
> > Thanks the authors for the detailed answers and also for incorporating an improved evaluation metric. I do appreciate the effort the authors took to improve the paper. I'm willing to increase my score from 6 to 7.
> >
> > One additional question: Can the authors give a quantitative or qualitative analysis on how the behaviors of the policies differ? As we discussed, it is possible that the policies alternate between a few behaviors. If this is the case, future work could use this insight to further improve the algorithm.

---

> ### Author Response · Authors · 2024-08-12
> **Thanks for the response**
>
> Thank you for your response. We are delighted to hear that our rebuttal addressed your concerns well.
>
> To address the issue you mentioned, even though it is unlikely to occur, we need to quantify the behaviors of each policy or measure the divergence among more than two policies [1]. However, quantifying the policy behaviors is challenging due to the difficulty of designing a best behavior representation method. Recent works share insights for behavior representation, such as trajectory distributions [2] and state-action pairs [3]. We will investigate the efficient representation approach and choose a more powerful divergence [1] for our work in the future.
>
>
> [1] Lu Mingfei, et al., "Measuring generalized divergence for multiple distributions with application to deep clustering", Pattern Recognition, 2024.
>
> [2] Dhruva Tirumala, et al., "Behavior Priors for Efficient Reinforcement Learning", JMLR, 2022.
>
> [3] Huang Zhiyu, et al., "Efficient Deep Reinforcement Learning With Imitative Expert Priors for Autonomous Driving", IEEE TNNLS, 2023.

---

### Official Review · Reviewer_jcAK · 2024-07-13

**Soundness:** 3
**Presentation:** 3
**Contribution:** 3
**Rating:** 7
**Confidence:** 4

**Summary:**

This paper is situated in the problem setting of heterogeneous cooperative agents, under the sequential update framework. The paper introduces the novel MADPO algorithm, in which agents maximize the Cauchy Schwarz divergence between agents and between episodes of data gathered by the same agent, to improve exploration. Empirical validation is performed on the Multi-Agent Mujoco and Bi-DexHands benchmark suites, demonstrating that the MADPO outperforms baselines.

**Strengths:**

Overall, the paper is clear, succinct, and the main idea is clear and easy to understand. The format, and figures are good, with all expected components included. The idea of maximizing the inter/intra agent divergences is intuitively appealing. Further, the authors address the pitfalls of naively maximizing intra-agent divergences by adopting the Cauchy Schwarz divergence. It's especially nice that maximizing the CS divergence implies maximizing the policy entropy as well. Experiments are done on a large number of tasks, with comparisons against expected baselines and parameter sensitivity analyses all present.

**Weaknesses:**

1. The motivation of the paper is not altogether clear to me. The paper seems to suggest that exploration is more challenging in the sequential update setting, necessitating devoted algorithms. Why would this be the case?
2. In many of the presented domains, the improvement of MADPO over the next best method is not very large. Sometimes, confidence intervals of MADPO overlap those of the next best method. Can the authors provide statistical significance tests for the main results in Figures 2 and 3, comparing MADPO to the next best method?
3. Some minor suggestions:
- Please check your paper carefully for typos, as there are quite a few:
    - Line 89: "connecting link dimension curse"? Not sure what this is
    - No period after Figure 4
    - Trust interval -> confidence interval
    - Lacking 'and' at line 174
    - Line 204: conditoned -> conditioned
    - Line 216: extra "of"
- Please be sure to state the number of trials in the main text. It is mentioned in the Neurips checklist, but I could not find it  in  the main text
- Please make the colors of the methods the same for both domains (i.e. pick 1 color for MADPO and be consistent with it)

**Questions:**

1. How sensitive is the algorithm to the scale of the divergence rewards? Have you done a study on this?
2. On the intra-policy divergence: is the policy updated every episode? If not, then wouldn't the intra-policy divergence reward often be 0?
3. Line 180 states that it would be challenging to define an inter-agent divergence in the simultaneous update scheme. Why not consider the divergence between $\pi^i_k$ and $\pi^j_{k-1}$? But this does not seem any more challenging to compute, and can be computed under  CTDE assumptions.
4. Would it be possible to implement this exploration scheme in the CTDE setting? If so, it would be interesting to see how well the method performs.
5.   Proposition 2 states that the CS divergence has a lower bound. Does it also have an upper bound?

**Limitations:**

yes

---

> ### Author Rebuttal · Authors · 2024-08-05
>
> First, we would like to express our gratitude for your careful review of our work, as well as for your positive comments and insightful suggestions.We address your concerns as follows:
>
> ## (1) Motivation of our work
>
> The sequential updating scheme offers a novel solution to heterogeneous MARL, enabling agents to access information from preceding ones.
>
> Our focus is on designing a sequential exploration strategy within this framework. Simply applying simultaneous exploration approaches, such as entropy or intra-agent divergence, to sequential methods may not fully leverage the available sequential information.
>
> Besides, in heterogeneous tasks, exploration should not only aim for novel state and policy discovery but also enhance heterogeneity. Existing sequential methods often disable parameter sharing to accommodate heterogeneity, which is a passive approach lacking active guidance.
>
> Therefore, in this work, we propose maximizing mutual policy divergence to actively enhance exploration and heterogeneity.
>
> ## (2) Statistical significance tests
>
> We believe that results presented with confidence intervals are relatively convincing in terms of statistical significance. To further evaluate the overall performance of MADPO, we utilize ***rliable***, a powerful statistical testing tool recommended by Reviewer 3yBc. We choose the aggregate interquartile mean (IQM) across multiple tasks as the metric for the sample efficiency test. Please refer to the author rebuttal and the attached PDF for more information.
>
> Figure (1) in the attached PDF shows that MADPO outperforms other baselines in terms of sample efficiency with a significant improvement gap in most tasks, and achieves the highest best episodic rewards in all tasks consistently.
>
> ## (3) Minor suggestions
>
> Thank you for pointing out the typos in our paper; we have corrected them accordingly. We have also adjusted the figures to highlight our method in red and included all experimental details in the main text.
>
> Regarding the statements in Line 89, we would like to clarify that in MATRPO [1], proposed by Li and He, a communication method using *connecting links* is proposed for sharing information among agents. However, as the number of agents increases, the additional cost of these connecting links cannot be ignored.
>
> ## (4) Sensitivity to the scale of divergence reward
>
> In CS divergence, the kernel width $\sigma$ controls the scale of divergence. We conducted parameter sensitivity experiments, as presented in the original manuscript, which demonstrate that MADPO is somewhat sensitive to $\sigma$. Additionally, we performed an IQM test to further investigate this sensitivity in the rebuttal. Please refer to the author rebuttal and the attached PDF for more information.
>
> We can observe from Figure 2 (c) that, even though MADPO is slightly sensitive to $\sigma$, it outperforms HAPPO in a reasonable range of $\sigma$ across several tasks. Note that we show here the aggregate results of parameter sensitivity experiments, and we can tune $\sigma$ in each tasks individually for further improvement.
>
> ## (5) Policy updating in intra-agent divergence maximization
> The policy is updated every episode. The intra-agent policy divergence measures the difference between one agent’s current policy and its policy in the last episode. Thus, except for the first episode, once the agent’s policy is improving, the intra-agent policy divergence would not be zero.  And the policy is updating according to the divergence.
>
> ## (6) Concerns about CTDE
> First, we would like to clarify that our MADPO is based on CTDE. Because MADPO has a global V network and multiple agent policy networks and follows CTDE settings. The centralized V network generates a V function for training agent networks, while decentralized agents interact with the environment individually, as indicated in Algorithm 1 of the original manuscript.
>
> Computing the divergence between $\pi_k^i$ and $\pi_{k-1}^j$ is not practically challenging indeed. However, diversifying agents in simultaneous updating methods, such as MAPPO, requires the non-parameter sharing setting, under which MAPPO will lose the monotonic improvement guarantee, even though it may bring better performance [2]. On the contrary, in sequential methods, such as HAPPO and our MADPO, the objective naturally takes the preceding information into account, and maintains the monotonic improvement guarantee in heterogeneous tasks.
>
> Therefore, we made the statement that adapting the inter-agent divergence maximization to simultaneous updating methods is theoretically challenging. Additionally, we are also aware that adapting the proposed model to value-based methods, such as QMIX, is feasible, since it does not have the monotonic improvement issue.
>
> ## (7) Upper bound of CS divergence
> The CS divergence has an upper bound. According to Proposition 1 in the original manuscript, the upper bound of CS divergence $D_{CS} (\pi||\overline{\pi})$ is the sum of two 2-order Rényi entropies: $$D_{CS} (\pi||\overline{\pi}) \leq  \frac{1}{2}\mathcal{H}_2(\overline{\pi})+\frac{1}{2}\mathcal{H}_2(\pi).$$ Furthermore, given a finite action set $\pmb{A}=\\{a_0,...,a_n\\}$, the entropy $\mathcal{H}_2(\pi)$ has an upper bound $\log(n)$ when $\pi$ is a uniform distribution. Thus, the CS divergence is upper bounded by $\log(n)$, where $n$ is the number of the actions.
>
> [1] Li, Hepeng, and Haibo He., "Multiagent trust region policy optimization.", IEEE TNNLS 2023.
>
> [2] Kuba, J. G., et al., "Trust Region Policy Optimisation in Multi-Agent Reinforcement Learning.", ICLR 2022.

---

> > ### Comment · Reviewer_jcAK · 2024-08-11
> >
> > I have read the common response and the rebuttal to my specific review. Thanks for going over my review in detail. I am satisfied with the rebuttal to most of my points, except for point 6 (see below). The rebuttal underlines my belief that this is a good paper which should be accepted. However, my score already reflects this belief, so I will not change it.
> >
> > Recent work by (Sun et al. AAMAS 23)[https://arxiv.org/pdf/2202.00082] shows that even decentralized IPPO maintains the monotonic improvement guarantees by maintaining an approximate trust region, so I disagree that MAPPO w/o PS would lose that guarantee. It would be interesting if the authors could show that their proposed exploration technique works for the simultaneous decision-making approaches as well. However, this is an auxiliary point that is not directly relevant to the acceptance of this paper.

---

> ### Author Response · Authors · 2024-08-12
> **Thanks for the response**
>
> Thank you for your response. We are delighted to hear that our rebuttal addressed your concerns well.
>
> We agree that Sun et al. [1] did great work and offered a monotonic improvement guarantee for decentralized agents in MAPPO and IPPO. They ensured the trust region optimization by bounding the independent ratio. However, we would like to highlight that, decentralized agents are not equal to hetergeneous agents. In [1], when bounding the ratio, the authors noted the  necessity for paremeter sharing (please see the statement above Eq. 16 in [1]). Thus, when switching off the paremeter sharing, it remians unclear whether the guarantee proposed in [1] holds.
>
>
> [1] Sun Mingfei, et al., "Trust Region Bounds for Decentralized PPO Under Non-stationarity", AAMAS, 2023.

---

### Official Review · Reviewer_sHA3 · 2024-07-14

**Soundness:** 3
**Presentation:** 3
**Contribution:** 2
**Rating:** 5
**Confidence:** 4

**Summary:**

This paper introduces a novel multi-agent reinforcement learning (MARL) method called Multi-Agent Divergence Policy Optimization (MADPO), which enhances exploration and heterogeneity through a mutual policy divergence maximization framework. MADPO leverages a sequential updating scheme and quantifies discrepancies between episodes and agents, termed intra-agent divergence and inter-agent divergence, respectively. To address the instability and lack of directionality in traditional divergence measurements, the paper proposes using conditional Cauchy-Schwarz divergence to provide entropy-guided exploration incentives. Experiments demonstrate that the proposed method outperforms state-of-the-art sequential updating approaches in two challenging multi-agent tasks with various heterogeneous scenarios.

**Strengths:**

1. **Innovation**: The paper introduces MADPO, a novel MARL method that enhances agent exploration and heterogeneity through mutual policy divergence maximization.

2. **Theoretical Foundation**: The use of conditional Cauchy-Schwarz divergence to address instability and directionality in traditional divergence measurements is a contribution.

3. **Experimental Validation**: The experiments conducted on two challenging multi-agent tasks with different heterogeneous scenarios convincingly demonstrate the effectiveness and superiority of MADPO in enhancing exploration and heterogeneity.

**Weaknesses:**

1.  The paper lacks analysis and comparison with relevant literature on sequential decision-making, such as:
   - Liu J, Zhong Y, Hu S, et al. Maximum Entropy Heterogeneous-Agent Reinforcement Learning[C]//The Twelfth International Conference on Learning Representations.  (This paper extends SAC to heterogeneous sequential decision-making scenarios, and the relationship between this work and the current paper remains unclear.)

2. It is unclear whether the intrinsic reward method proposed in this paper can ensure that the resulting trained policies are consistent with the original policies.

**Questions:**

1. Could you provide a detailed comparison between your proposed MADPO method and the approach presented in "Maximum Entropy Heterogeneous-Agent Reinforcement Learning" by Liu et al.? Specifically, how does MADPO improve upon or differ from this method in terms of handling heterogeneous sequential decision-making scenarios?

2. I may have missed some details, but could you clarify whether the intrinsic reward method in MADPO ensures that the trained policies remain consistent with those optimized solely based on the original rewards?

**Limitations:**

The authors have addressed the limitations of their work and discussed potential negative societal impacts in accordance with the guidelines.

---

> ### Author Rebuttal · Authors · 2024-08-05
>
> Thanks for your positive suggestions. We address your concerns as follows:
> ## (1) Comparison with HASAC
> Liu et al. proposed Heterogeneous Agent SAC (HASAC) by extending maximum entropy RL into heterogeneous MARL [1]. However, we would like to clarify that our MADPO is an on-policy MARL method, while HASAC is based on an off-policy manner. Thus, HASAC naturally enjoys higher sample efficiency than other on-policy sequential updating methods, such as HAPPO, HATRPO and A2PO. However, as an off-policy method, HASAC incurs significant time costs.
>
> In terms of handling heterogenous tasks, HASAC shares the similar idea in HAPPO, decomposing the joint Q function (joint advantage function in HAPPO) to individual ones conditioned by other agents’ policies. Nevertheless, HASAC lacks an active optimization objective to guide agents toward greater heterogeneity and diversity. In terms of enhancing exploration, HASAC adopts the strategy of SAC by incorporating a policy entropy term into the reward. In contrast, MADPO actively maximizes the stable mutual divergence of policies while implicitly maximizing the policy entropy, providing a stronger incentive for exploration and heterogeneity, as presented in Section 4 of the original manuscript.
>
> To comprehensively compare MADPO with other sequential updating approaches, we conducted overall performance and wall time experiments for HASAC and our method. Please refer to the author rebuttal and the attached PDF for more information. We can observe from Table 1 and Figure 4 in the attached PDF that, although HASAC achieves a performance improvement, its huge time costs cannot be ignored.
>
> Additionally, we recognize that the proposed mutual policy divergence maximization can be adapted to off-policy methods as well, which is our future research direction.
>
> ## (2) Consistency between policies trained on MADPO and the original reward
>
> As an intrinsic reward, mutual PDM in MADPO aims to apply incentives when the change of extrinsic reward extrinsic rewards does not lead to policy improvements. Therefore, in MADPO, when the episode reward is increasing, the original reward part is dominant, resulting in policies that keep consistency with those trained on extrinsic rewards. On the other hand, when the reward struggles to improve, indicating agents should explore the environment, the intrinsic reward provides guidance to encourage agents to escape from suboptimal policy. In such cases, the policy generated may deviate from the original policy. To tackle this issue, we use parameter $\sigma$ to control the scale of the intrinsic reward, ensuring consistency between policies trained on the original and proposed rewards.
>
> We have conducted the experimental comparison with the $no$ $incentive$ setting. We choose here the interquartile mean (IQM) across tasks as the metric. Please refer to Figure 2 of the attached PDF in author rebuttal above. The results indicate that mutual PDM in MADPO leads to significant policy improvements over the original reward. We have also conducted the parameter sensitivity experiments for $\sigma$, revealing that the mutual PDM with a reasonable choice of $\sigma$ will not cause performance degradation.
>
> [1]  Liu, Jiarong, et al., "Maximum Entropy Heterogeneous-Agent Reinforcement Learning", ICLR 2024.

---

### Author Rebuttal · Authors · 2024-08-05

We thank all reviewers for their encouraging comments and constructive feedback. We are glad to note that the reviewers recognized our work as innovative, appealing and easy-to-follow *[sHA3, jcAK, 3yBc, gpQ4]*, theoretically nice and interesting to RL community *[sHA3, jcAK, 3yBc, gpQ4]*, well-organized and well-written *[jcAK, 3yBc, gpQ4]*, and experimentally effective *[sHA3, gpQ4]*.

We report here the additional comparison to address several concerns regarding the experimental results.
We re-evaluate our method with baselines by using ***rliable*** [1], a powerful statistical testing toolkit suggested by Reviewer 3yBc. Since the environments we used in this work do not have a round end score, we choose the aggregate interquartile mean (IQM) sample efficiency test of ***rliable*** for evaluation.

First, we compare the IQM rewards of our method across multiple tasks against other MARL baselines *[jcAK, 3yBc]*, against other exploration incentives *[gpQ4]*, and under different parameter settings *[jcAK, 3yBc]*. Then, we compare our method with the SOTA off-policy sequential method HASAC, in terms of overall performance and running time *[sHA3]*. Lastly, we present the running time results of MADPO and other baselines *[gpQ4]*. All the additional results are included in the attached PDF.

## Aggregate IQM sample efficiency test
The interquartile mean (IQM) computes the mean scores of the middle 50% runs, while discarding the bottom and top 25%. Here, we evaluate the performance across multiple tasks, and the total number of runs is $n \times m$, where $n$ is the number of trials for one task ($n=5$ in this paper), $m$ is the number of tasks. IQM test is more robust than the mean and has less bias than the median. The lines in the figures represent the IQM, while the shaded areas indicate the confidence intervals.

## Aggregate IQM comparison with MARL baselines *[jcAK, 3yBc]*
Figure 1 shows the IQM rewards comparison against baselines across $10$ tasks in Bi-Dexhands and $9$ tasks in MA-Mujoco. The $10$ tasks in Bi-Dexhands and include all the Bi-Dexhands tasks used, and the $10$ tasks in MA-Mujoco include all the MA-Mujoco tasks used, in the original manuscripts. The $3$ tasks of MA-Mujoco Ant include *Ant-v2-2x4*, *Ant-v2-4x2*, and *Ant-v2-8x1*. The $3$ tasks of MA-Mujoco Halfcheetah include *Halfcheetah-v2-2x3*, *Halfcheetah-v2-3x2*, and *Halfcheetah-v2-6x1*. The $3$ tasks of MA-Mujoco Walker2d include *Walker2d-v2-2x3*, *Walker2d-v2-3x2*, and *Walker2d-v2-6x1*.

We observed that the proposed MADPO consistently outperforms the state-of-the-art MARL methods in terms of best episodic reward across multiple tasks. The results also show that, MADPO has higher sample efficiency compared to other methods and achieves an improvement gap in most tasks.

## Aggregate IQM comparison with other exploration incentives *[gpQ4]*
Figure 2 (a) and (b) show the IQM rewards comparison against other exploration incentives across $10$ tasks in Bi-Dexhands and $10$ tasks in MA-Mujoco. We observed that in MA-Mujoco, the CS-divergence outperforms other incentives with a small overlap and narrow confidence interval, indicating its better stability than KL-divergence. In $10$ tasks of  Bi-Dexhands, Figure 2 (b) shows that MADPO is the only one that keeps increasing with a significant improvement gap after 10m steps.

## Aggregate IQM comparison under different parameter settings *[jcAK, 3yBc]*
Figure 2 (c) and (d) indicate the parameter sensitivity of MADPO across $10$ tasks in Bi-Dexhands and $10$ tasks in MA-Mujoco. We can observe that MADPO is a little sensitive to parameter $\sigma$. However, in terms of IQM, it still outperforms HAPPO in a reasonable range of $\sigma$. Note that we can individually tune $\sigma$ for each task for further performance improvement. Figure 2 (d) indicates that MAPDO is also a little sensitive to the parameter $\lambda$, yet it consistently shows better performance than HAPPO. In conclusion, we suggest that when tuning $\sigma$, frequently used values can be a good choice, and tuning $\lambda$ should consider the characteristics of the task, and do not set $\lambda$ more than 0.5 in most cases.

## The performance and the running time comparison against on-policy baselines and HASAC *[sHA3, gpQ4]*
Here we compare MADPO with off-policy baseline HASAC on five random seeds and two tasks of Bi-Dexhands and two tasks of MA-Mujoco (due to the limited time), as indicated in Figure 3. We can observe that the off-policy HASAC has a performance improvement compared to on-policy methods. However, in Table 1, the running time of HASAC is much more than on-policy methods, which is a trade-off between time cost and performance. Table 1 also indicates that compared to baselines, MADPO only introduces a negligible extra time cost.

[1]  Agarwal, Rishabh, et al., "Deep reinforcement learning at the edge of the statistical precipice.", NeurIPS 2021.

---

### Decision · Program_Chairs · 2024-09-25

**Decision:**

Accept (poster)

**Comment:**

The paper presents Multi-Agent Divergence Policy Optimization (MADPO), which introduces a mutual policy divergence maximization framework aimed at enhancing exploration and heterogeneity in MARL. The method leverages a sequential updating scheme, allowing agents to learn from preceding ones, and proposes the use of conditional Cauchy-Schwarz (CS) divergence to address the stability and directionality issues in traditional divergence measurements. The paper is well-grounded in theory, with clear explanations of how the proposed method addresses these stability and directionality challenges in policy divergence measurements. The results in heterogeneous multi-agent environments demonstrate that MADPO outperforms existing methods, providing strong empirical evidence for the effectiveness of the proposed approach. The reviewers are satisfied with the addition of statistical significance tests, as well as comparisons to related works and other baselines. Based on this, the paper is recommended for acceptance.